# Stratospheric trace gas profile retrievals from balloon-borne limb imaging of mid-infrared emission spectra

Ethan Runge[1], Jeff Langille[2], Daniel Zawada[1], Adam Bourassa[1], and Doug Degenstein[1]

[1]University of Saskatchewan, Saskatoon SK, Canada
[2]University of New Brunswick, Fredericton NB, Canada

**Correspondence:** Ethan Runge (ethan.runge@usask.ca)

**Abstract.** The Limb Imaging Fourier-Transform Spectrometer Experiment (LIFE) instrument is a balloon-borne prototype of a satellite instrument designed to take vertical images of atmospheric limb emission spectra in the 700 to 1400 $cm^{-1}$ wavenumber range from the upper troposphere/lower stratosphere (UTLS) altitude region of the atmosphere. The prototype builds on the success of past and existing instruments while reducing the complexity of the imaging design. This paper details the results of a demonstration flight on a stabilized stratospheric balloon gondola from Timmins, Canada in August 2019. Retrievals of vertical trace gas profiles for the important greenhouse gases $H_2O$, $O_3$, $CH_4$ and $N_2O$, as well as $HNO_3$, are performed using an Optimal Estimation Approach and the SASKTRAN radiative transfer model. The retrieved profiles are compared to approximately coincident observations made by the ACE-FTS solar occultation and MLS instruments. An evaluation of the LIFE measurements is performed and areas of improvement are identified. This work increases the overall technical readiness of the approach for future balloon, aircraft and space applications.

## 1 Introduction

The Limb Imaging Fourier-Transform Spectrometer Experiment (LIFE) is an imaging Fourier transform spectrometer (IFTS) developed by the University of Saskatchewan in collaboration with ABB Canada and with funding from the Canadian Space Agency. The eventual goal is to use vertical imaging from low earth orbit to obtain high spatial resolution atmospheric composition measurements in the upper troposphere/lower stratosphere (UTLS) region of the atmosphere (Runge et al., 2021). The current prototype uses a linear array detector and imaging optics designed to cover the limb altitudes from typical float altitude of a stratospheric balloon platform. The prototype is unique in design as it makes use of a modified commercially available interferometer, forgoes the use of fore-optics, and has only one cooled element in the form of a cold-stop assembly to mitigate the impact of instrument self-emission rather than using fully cooled optics. The first technology demonstration flight of the instrument was performed from a stratospheric balloon launched and operated by Centre National d'Etudes Spatiales (CNES) from Timmins, Canada in August of 2019. The development, characterization and calibration of the instrument is detailed in-depth in an earlier publication (Runge et al., 2021). The current work details the approach to retrievals of trace gas profiles, and the results are characterized in terms of precision and resolution.

The prototype draws on several decades worth of community research and advancement in the field of FTS instruments as a method of remote sensing for atmospheric trace gases. The primary influence on the LIFE instrument is the Gimballed Limb Observer for Radiance Imaging of the Atmosphere (GLORIA). GLORIA is an IFTS designed for aircraft and balloon platforms and has very similar goals and design elements to LIFE (Riese et al., 2014; Friedl-Vallon et al., 2014). Prior to the development of LIFE, GLORIA had been validated using measurements from several campaigns and produced rich datasets indicating the robustness of the concept (Johansson et al., 2018; Sha, 2013; Kleinert et al., 2014; Olschewski et al., 2013; Monte et al., 2014; Kaufmann et al., 2015; Piesch et al., 2015; Ungermann et al., 2015). The goal of the LIFE prototype was to build on these successes and create an instrument of similar capabilities with less complexity. The spectra obtained from the LIFE demonstration flight indicate that design specification goals were met, though residual issues with radiometric calibration remain (Runge et al., 2021).

Critical heritage for these instruments comes from the Michelson Interferometer for Passive Atmospheric Sounding (MI-PAS), the forebear of GLORIA. While MIPAS is not an imaging instrument, i.e. it uses a single element detector instead of an array, many aspects of the design are applicable to both GLORIA and LIFE (Fischer et al., 2008; Friedl-Vallon et al., 2004; Fischer and Oelhaf, 1996). The approach to trace gas retrievals from the MIPAS reduced resolution mode provide valuable insight into the choice of spectral windows used for the LIFE retrievals (Fischer and Oelhaf, 1996; von Clarmann et al., 2009).

In this study we also use measurements from the Atmospheric Chemistry Experiment FTS (ACE-FTS) and Microwave Limb Sounder (MLS) instruments. ACE-FTS has been operational onboard SciSAT for nearly two decades and provides limb measurements at dusk and dawn via solar occultation (Bernath et al., 2005). Measurements made by ACE have been validated against other satellite instruments, ground-based and aircraft-based sources and are generally accepted as accurate (Walker et al., 2005; Waymark et al., 2014; Walker et al., 2018, 2021). Each of the target species we retrieve here with LIFE have been validated for ACE-FTS: $H_2O$ (Weaver et al., 2019; Davis et al., 2021; Sheese et al., 2017), $O_3$ (Bognar et al., 2019; Sheese et al., 2022, 2017), $N_2O$ (Sheese et al., 2017; Plieninger et al., 2016) and $CH_4$ (Plieninger et al., 2016). Additional species like $HNO_3$ have also been validated (Sheese et al., 2017). Due to the high coverage and availability of data at the time of the Timmins flight, trace gas profiles determined from ACE measurements provide a good benchmark by which to judge the validity of the profiles returned by the LIFE data analysis. Additional coverage and comparison is provided by measurements taken from MLS, which makes use of heterodyne radiometers measuring in the thermal regime (Waters et al., 2006). Similar to LIFE, GLORIA and MIPAS, thermal emission measurements allow MLS to take measurements during both day and night without the need for external sources, greatly increasing the potential coverage (Waters et al., 2006). The MLS measurements have been validated for all target species except $CH_4$, which it does not measure: $H_2O$ (Lambert et al., 2007; Vömel et al., 2007; Barnes et al., 2008; Hurst et al., 2014), $O_3$ (Jiang et al., 2007; Froidevaux et al., 2008), $N_2O$ (Lambert et al., 2007) and $HNO_3$ as well (Santee et al., 2007). A comparison to several ACE and MLS profiles taken close to the day of the LIFE flight are included as a verification that the LIFE prototype meets the scientific goals in addition to the design specifications indicated in the prior publication (Runge et al., 2021).

In this paper, the retrieval algorithm used to determine the trace gas profiles is first described in a general sense. A more detailed accounting of how the general algorithm is modified for the particular aspects of the LIFE instrument follows, along

with a description of the trace gas profile determination for each of the species of interest. Following the individual description of the retrieved quantities, a direct comparison of generally accepted profiles obtained from ACE measurements is done to demonstrate the quality of the LIFE profiles, verifying that the prototype concept achieves the scientific goals. These results show the validity of utilizing commercially available off-the-shelf components and simpler optics for reduced cost, and un-cooled optics for reduced complexity, marking attractive options for future space-based instruments.

## 2 LIFE Instrument

The LIFE instrument is made with several custom-made anodized aluminum cases containing the optics, control electronics and blackbodies. The optical path starts with a pointing system that can be aimed to view the atmospheric limb through one port or at either of the on-board blackbodies through other ports. The interferometer at the core of the instrument is a modified version of the commercially available MB3000 interferometer manufactured by ABB Canada. The aft-optics are custom made lenses, and focus the system on a 16 element mercury-cadmium telluride (MCT) detector, with a cold-stop cooled by an onboard sterling cooler to reduce self-emission signal.

The full field of view (FOV) of the instrument is 6.01 degrees, with each pixel covering 0.32 degrees. At the in-flight float altitude of 40 km, this FOV resolution results in a tangent altitude spacing of about 4 or 5 km between pixels. An interferogram is generated for each pixel every 2.26 seconds as the interferometer moves through the full -1.4 cm to 1.4 cm optical path difference (OPD) range. The OPD leads to a spectral sampling of 0.357 cm$^{-1}$ over the spectral range, 700 to 1400 cm$^{-1}$, for which LIFE measurements are taken.

Retrieval performance requires a noise equivalent spectral radiance (NESR) of 15 nW/cm$^2$/sr/cm$^{-1}$ (Runge et al., 2021). Individual scans have a higher noise than this, and so time-averaging is performed. For the measurements presented in this paper, 20 scans are averaged together, reducing the temporal cadence of the instrument to 45 seconds. The full characterization and description of the instrument is detailed in the LIFE instrument paper by Runge et al. (2021).

## 3 Retrieval

### 3.1 Overview

The retrieval method applied to the LIFE data follows the atmospheric inverse problem solution described in Rodgers (2000) and uses similar notation. The goal of this approach is to take a given set of measurement information, the measurement vector $\mathbf{y}$ with length $m$ that contains the instrument spectra obtained from the instrument, and determine a set of atmospheric state parameters, the state vector containing the atmospheric concentrations to be retrieved and other instrument parameters, $\mathbf{x}$ of length $n$, that determine the state of the atmosphere. The state vector is input to a forward model, $F$, that represents the physics of the atmosphere and the instrument to obtain a set of expected radiances. The inverse method is then concerned with iteratively changing the state vector until the forward model output matches the measurement vector. The iterative approach

determines how the state vector should be changed via minimization of the cost function

$$\chi^2 = (\mathbf{y} - F(\mathbf{x}))^T \mathbf{S}_\epsilon^{-1} (\mathbf{y} - F(\mathbf{x})) + (\mathbf{x} - \mathbf{x}_a)^T \Gamma^T \Gamma (\mathbf{x} - \mathbf{x}_a), \tag{1}$$

where $\mathbf{S}_\epsilon$ is the error covariance matrix associated with the measurement vector, $\mathbf{x}_a$ is the apriori state vector, and $\Gamma$ is a regularization matrix that determines the influence of the apriori state on the minimization. The elements of $\mathbf{S}_\epsilon$ are calculated from the measurement noise after spectral calibration. Using the Gauss-Newton minimization method leads to the iterative step formula

$$\mathbf{x}_{i+1} = \mathbf{x}_i + (\mathbf{K}_i^T \mathbf{S}_\epsilon^{-1} \mathbf{K}_i + \Gamma^T \Gamma + \gamma_i \mathrm{diag}(\mathbf{K}^T \mathbf{S}_\epsilon^{-1} \mathbf{K}))^{-1} [\mathbf{K}_i^T \mathbf{S}_\epsilon^{-1} (\mathbf{y} - F(\mathbf{x}_i)) - \Gamma^T \Gamma (\mathbf{x}_i - \mathbf{x}_a)], \tag{2}$$

with $\mathbf{K}$ representing the Jacobian matrix of the forward model, $i$ the step number and $\gamma$ a Levenberg-Marquardt damping factor which aids in the convergence of the Gauss-Newton method in non-linear cases (Fletcher, 1971).

Also of importance in the discussion of the retrieval is the averaging kernel, as it gives an indication as to the effective vertical resolution of the retrieved profile. The averaging kernel is defined as

$$\mathbf{A} = (\mathbf{K}^T \mathbf{S}_\epsilon^{-1} \mathbf{K} + \mathbf{S}_a^{-1}) \mathbf{K}^T \mathbf{S}_\epsilon^{-1} \mathbf{K}, \tag{3}$$

with the newly defined $\mathbf{S}_a^{-1} = \Gamma^T \Gamma$. The averaging kernel shows how each retrieval grid point depends on its neighbours. As such, the full-width at half-max (FWHM) for a given row of the averaging kernel is an indication of the vertical resolution of the retrieval. This can be modified by regularization factors, discussed further in Section 3.6.

Due to unique systematic artefacts within the LIFE measurement vector, the retrieval approach for atmospheric trace species needs to occur in a cycle of two or three retrievals, each solving for a different set of state vectors. The cycle required for each quantity of interest is discussed in the respective subsection of Section 4. The retrieval process applied at any of the stages is depicted by the flowchart in Fig. 1.

Each of the quantities sought uses the calibrated spectra and the results of the previously obtained quantity in the retrieval process. The flowchart depicted in Fig. 2 shows the general retrieval order. For each time-averaged set of measurements, the pointing is found first with the proper microwindow, which is followed by $H_2O$. The retrieved profile is input into the climatology, replacing the standard $H_2O$ profile for the next species retrieval, which is $O_3$. The process continues by replacing atmospheric profiles in the climatology with the retrieved profile; $HNO_3$ comes after $O_3$ and then $CH_4$ and $N_2O$. Also note in this schema that no retrieval of temperature is done due to uncertainty in radiometric calibration accuracy. The Modern-Era Retrospective analysis for Research and Applications (MERRA) temperature profile is used instead, which leads to some errors as discussed in Section 4.2.1.

## 3.2 Forward Model

The forward model consists of a radiative transfer model and an instrument model. The radiative transfer model portion makes use of the SASKTRAN radiative transfer framework. The original framework was developed at the University of Saskatchewan to provide efficient estimation of limb scatter measurements as a function of wavelength and tangent altitude (Bourassa et al.,

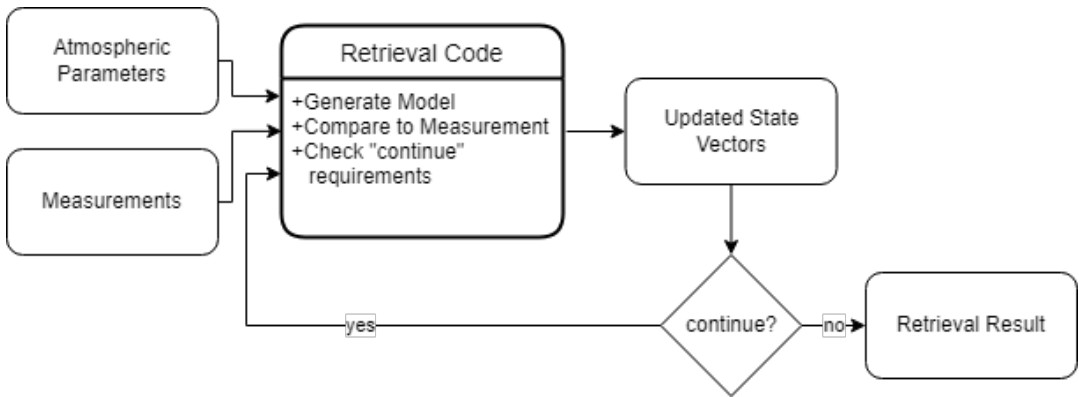

**Figure 1.** The general process used by the retrieval code. Inputs are given to the code as initial parameters, a comparison is made, and the parameters that the code has access to change (the state vectors) are updated. A set of requirements are provided on whether more iterations are needed. If more are needed, the updated state vector is used as the new input. If the completion criteria are met, the updated state vector output is the result.

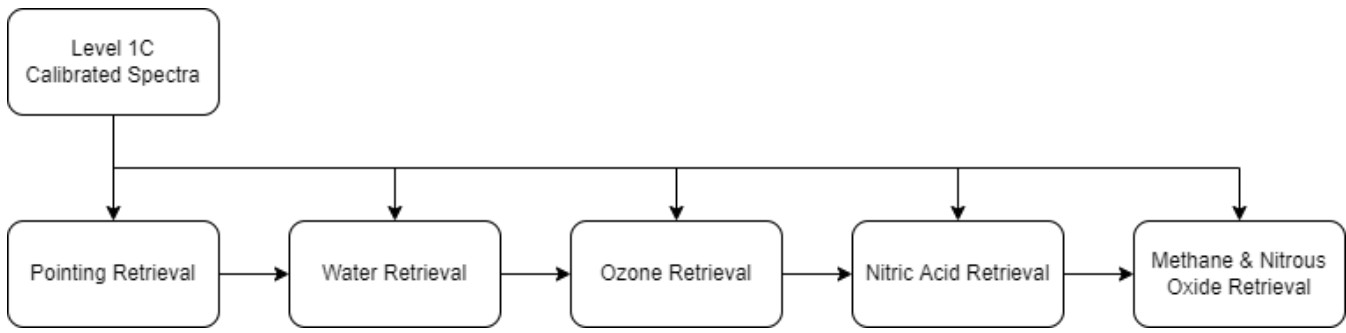

**Figure 2.** Broad flowchart for the determination of trace gas profiles applied to each scan of the LIFE data. Each of the stages is further divided into a retrieval process.

2008). The model uses a successive order approach to calculate measurements along rays traced through a spherical atmosphere. Since the initial inception, updates have been made to the framework to allow for the accurate calculation of high spatial resolution measurements in a non-spherically symmetric atmosphere, increasing the accuracy of the estimates (Zawada et al., 2015). This framework is referred to as SASKTRAN-HR. Further improvements to SASKTRAN-HR have been made that allow the analytic calculation of weighting functions, increasing the speed of calculations by many orders of magnitude (Zawada et al., 2017).

The most recent improvement to the SASKTRAN radiative transfer engine critical to the LIFE analysis is the development of a thermal radiative transfer model, known as SASKTRAN-TIR. The implementation of the thermal radiative transfer engine is detailed in Warnock (2020) and Jensen (2015). The framework breaks the atmosphere into a series of cells along the instrument lines of sight (LOS). Starting at the furthest point from the defined observer, the cell contributions are determined

and passed as an input to the next cell in the series. The total contribution of all cells, attenuated through each LOS is determined at the observer location. Spectral line intensities are calculated from the HIgh-resolution TRANsmission molecular absorption (HITRAN) database, utilizing the HITRAN2016 (Gordon et al., 2017) database for this implementation. The simulated environment implemented to generate high-resolution output is made with standard Fast Atmospheric Signature CODE (FASCODE) atmospheric constituents for most species (Anderson et al., 1986; Clough et al., 1987), SASKTRANs built-in Labow ozone profile and MERRA temperature and pressure profiles (Global Modeling and Assimilation Office (GMAO) (2015)). Voigt broadening profiles of the HITRAN spectral lines are calculated based on the provided MERRA temperature and pressure profiles.

The instrument model portion of the forward model takes the radiative transfer results and performs a convolution with the instrument line shape (ILS) based on instrument parameters. The LIFE instrument has 16 pixels with well known and consistent FOV values. The model is set up with an observer at a given height and LOS vectors. This information is also given to SASKTRAN such that the radiative transfer model generates measurements over the same LOS range as the LIFE measurements but at a much higher LOS density. Each pixel has an initial pointing vector, a calculated pitch correction, and a known non-zero FOV which define its light collection properties within the simulation. These three attributes work in conjunction with one another to create integration weights which are then applied to the high-density LOS model. This effectively performs a weighted average of the modelled signal for a given LOS and FOV in a manner that is representative of the physical operation of the LIFE instrument. The result is a high resolution spectrum for each of the LOS, an example of which is given in Fig. 6 b) as a representative radiance incident on to the instrument. These high resolution radiance spectra are used as input to the front end of the high fidelity instrument model that simulates real instrument effects such as the broadening due to instrument line shape.

## 3.3 Spectral windows

The retrieval spectral windows used in the LIFE retrievals are broadly based on microwindows used by von Clarmann et al. (2009) for MIPAS, since they are chosen for sensitivity in this spectral/altitude range, and optimal exclusion of non-local thermodynamic equilibrium (non-LTE) emissions. However, because of the relatively lower spectral resolution of the LIFE measurements we have often chosen broader ranges in this application. While admittedly not an optimal set in the sense of von Clarmann et al. (2009), these make a reasonable choice for demonstration of proof-of-concept retrievals. The windows used are given in Fig. 3. The left panel of the figure is a table indicating the species name and the wavenumber range with a colored background associated with that species. The right panel shows the full extent of the LIFE wavenumber range for each of the measurement LOS paths. The colored areas show the location of each of the corresponding species spectral windows in the larger spectral range.

## 3.4 State Vector

The state vector consists of state elements, individual parameters that the retrieval is allowed to modify in the retrieval process, which are used as input to the forward model. This typically includes the target species concentrations and instrumental factors.

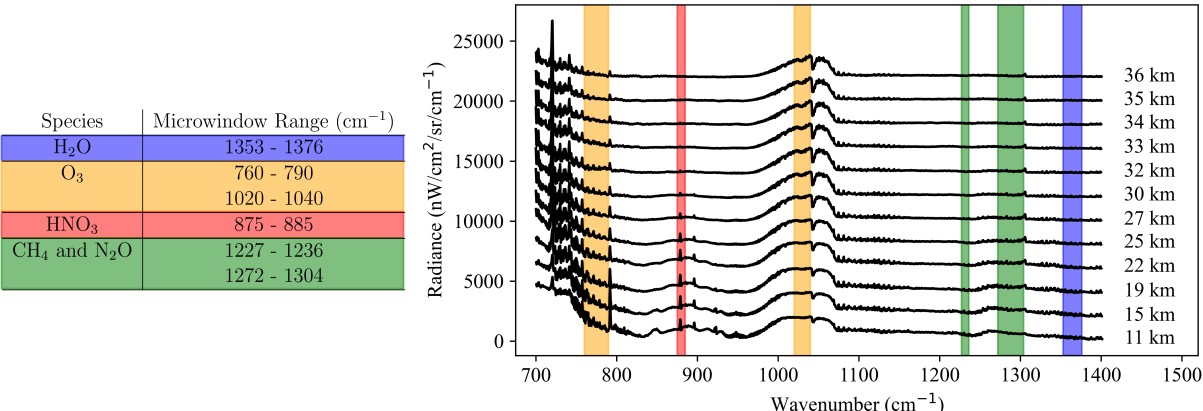

| Species | Microwindow Range (cm$^{-1}$) |
|---|---|
| H$_2$O | 1353 - 1376 |
| O$_3$ | 760 - 790 |
|  | 1020 - 1040 |
| HNO$_3$ | 875 - 885 |
| CH$_4$ and N$_2$O | 1227 - 1236 |
|  | 1272 - 1304 |

**Figure 3.** Spectral windows used in the LIFE retrievals. Each colored spectral span corresponds to the species with corresponding background color. Only 12 tangent heights are shown here for clarity; the bottom-most pixel LOS views the ground and is thus omitted, and the top three have tangent altitudes in the 36 km range and are omitted to prevent crowding.

In the case of target trace gas species profiles, the state vector contains concentrations in terms of parts per million by volume (ppmv) at discrete grid points with a spacing of 1000 m in the range covered by LIFE, approximately 11 km to 40 km. The choice of ppmv is made in this particular case since the profiles are smoother than number density retrievals.

LIFE takes simultaneous measurements along each LOS instead of using a scanning technique. This means that every LOS measurement is taken by a different pixel, each requiring special corrections for systematic biases and artefacts remaining in the radiometrically calibrated data (Runge et al., 2021). Each pixel has a series of state elements that are used to correct issues that may still exist after the radiometric calibration. These come in the form of a radiometric offset to shift the baseline, a radiometric slope to correct for systematics appearing in select microwindows and a shift of the spectra to accurately register wavenumber. These three correction parameters are unique for each of the microwindows used in the retrieval process. A final state element used for the LIFE measurements is a continuum profile that is used to correct for baseline shifts that cannot be corrected for with radiometric offsets or slopes.

### 3.5 Retrieval Order

The first step in obtaining a complete set of retrieved profiles is the determination of the instrument pointing. Figure 4 depicts the process for pointing determination, where the pointing parameter is the only kept value, used in every subsequent species retrieval from the same time period. The core section "Pointing Retrieval" follows the process from Fig. 1.

On-board telemetry and knowledge of the pointing mirror orientation allow a close determination of the instrument pointing, but even with care taken to choose a stable time period, the averaging in time required to meet noise constraints means that some error in the pointing is expected (Runge et al., 2021). In this case, an appropriate shift to the instrument pointing is

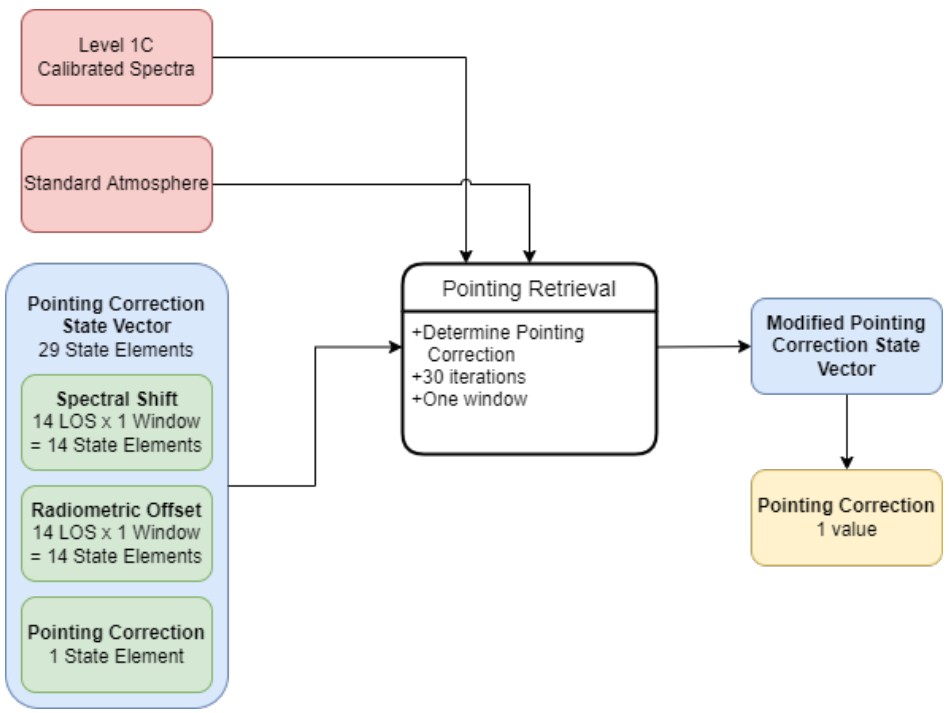

**Figure 4.** The pointing retrieval takes as input a standard atmosphere and state vectors allowing pointing, spectral shift and radiometric offset to change, and returns the pointing angle parameter associated with the best fit.

determined for the time period from which the measurements are taken. $CO_2$ windows are chosen for the pointing retrieval due to the trace gas being nearly homogeneous in the atmosphere and having little altitude dependence (von Clarmann et al., 2009).

Trace gas retrievals then follow, in the order of $H_2O$, $O_3$, $HNO_3$, and $CH_4$ joined with $N_2O$. A detailed breakdown of the water retrieval process for example is given in Fig. 5. Each of the "stage" blocks in this flowchart follow the process set out in Fig. 1.

The water retrieval first stage takes as input a standard atmosphere, the LIFE measurements, three initial correction state vectors and the previously determined pointing as a frozen parameter. These state vectors allow a wavenumber shift to align peaks, and both a constant and linear radiometric correction to remove residual radiometric correction errors in the baseline.
The correction state vector results are saved for use in the second stage, where a continuum correction is added as a variable. The initial states for stage two are the results from stage one, with the spectral shift vector no longer being allowed to change. In stage three, the spectral shift and continua are frozen, the previous stage two radiometric corrections are used as an initial guess that can be modified by the process, and the water profile is now allowed to change. At the end of this step, the water profile is determined. Similar processes to Fig. 5 are applied to $O_3$, $HNO_3$ and the co-retrieved $CH_4$ and $N_2O$, though $H_2O$ is
the most complicated. Section 4.2 describes more in-depth the state vectors and the necessity of each parameter.

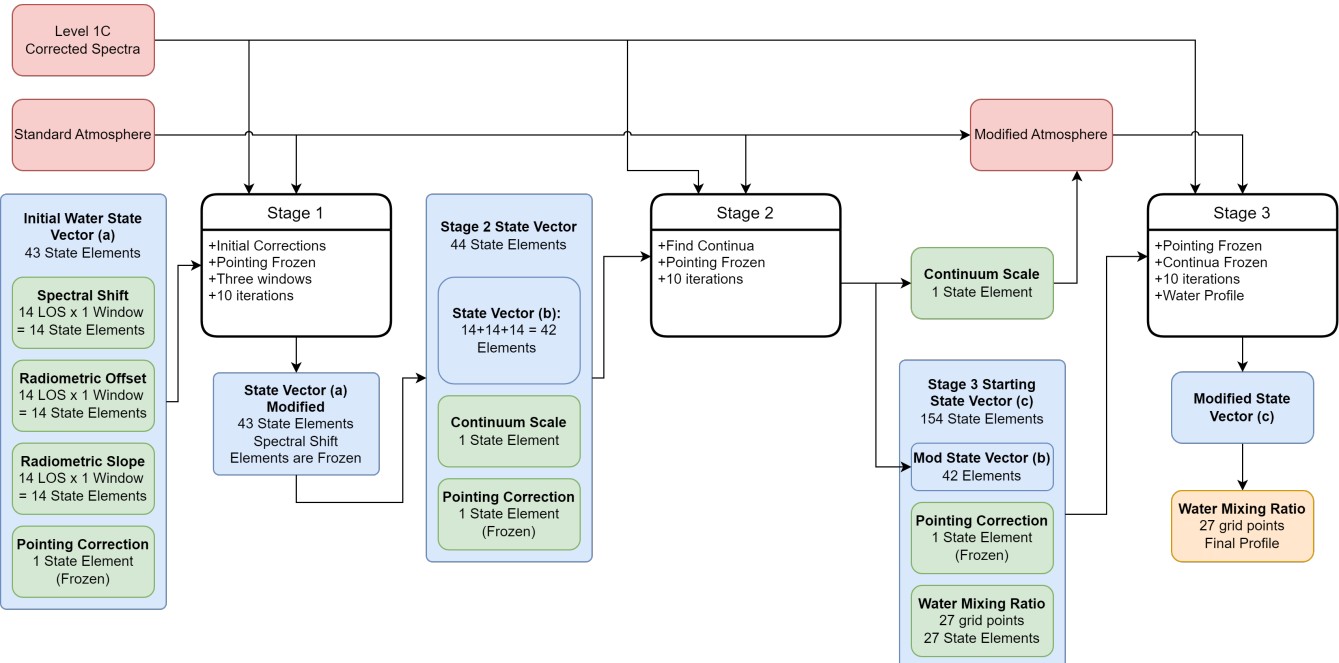

**Figure 5.** Breakdown of the $H_2O$ profile retrieval step for LIFE instrument scans. Red boxes indicate parameters set by methods other than the state vectors, blue boxes represent state vector parameters, and green boxes are user defined parameters. The orange box represents the final output of the retrieval process.

In general, the trace gas profile determined in the previous retrieval is used as an input in subsequent retrievals as shown in Fig. 2. Pointing is used as an input for $H_2O$, then pointing and the $H_2O$ profile are used for $O_3$, etc. In this way, major contributors to the thermal signal are determined first to minimize their ability to skew results for those species with smaller contributions.

## 3.6 Regularization

The vertical profile grid utilized in the LIFE retrievals is finer than the measurement grid, leading to large oscillations in the trace gas profile solutions unless regularization is applied. A second-order Tikhonov regularization is applied to the averaging kernel to smooth the profile by inducing a dependence in each point of the retrieved concentration profile on the neighboring points. This smoothing will also reduce the effective resolution of the retrieved profile to less than the initial retrieval grid. The effective resolution for each of the retrieved grid points is given by the FWHM of the averaging kernel associated with that grid point. For each of the retrieved species, a regularization factor determining how strong the dependence on neighboring points is chosen such that the FWHM is close to the measurement vertical resolution, about 5 km on average, when possible. Regularization above the measurement resolution begins to average information from different measurement vectors and results in

information loss with respect to the profile structure. Section 4.2.4 details the averaging kernel and resulting vertical resolution
for each of the species retrievals.

## 4 Retrieval Implementation and Results

### 4.1 Process and Pointing

To retrieve trace gas profiles from the LIFE measurements, the processing chain must begin with a correction to the LOS
pointing angles. The methodology and need for this correction is identified in von Clarmann et al. (2009) where $CO_2$ is used to
215 correct for the pointing of the MIPAS instrument. For the LIFE prototype we follow this example, using $CO_2$ as well, covering
most of the same range as was used in the MIPAS case, 745 cm$^{-1}$ to 765 cm$^{-1}$ as this is a window where the features are not
heavily obscured by other atmospheric species (von Clarmann et al., 2009). The correction to pointing information retrieved
from the flight gondola is deemed necessary due to uncertainties in the gondola stability over the time of measurements that
are averaged together and the consistency of the pointing apparatus employed by the instrument. Using a known quantity such
as $CO_2$, which is determined from observations made at the Mauna Loa National Oceanic and Atmospheric Administration
(NOAA) facility at the time of flight (408 parts per million (ppm)), allows for the retrieval to find the optimal pointing correction
(https://www.esrl.noaa.gov/gmd/ccgg/trends/monthly.html). A representative measurement made by one of the pixels after the
45 second time averaging is applied and is shown in Fig. 6 a).

As per the discussion of Section 3.2, the initial pointing vectors are generated from flight data retrieved from the gondola.
In Fig. 6 b) the forward model intermediate steps of determination of the high resolution output of the radiative transfer model
along an LOS and the ILS for the corresponding pixel it is to be convolved with are depicted. The blue line in Fig. 6 c) shows
the first spectrum of the retrieval process based on initial parameters. The orange line in this figure shows the result after one
iteration that is allowed to change correction parameters, which can be seen to be much closer to the measurement than the
prior attempt.

All LOS vectors are adjusted using the same correction factor and the state vector is updated, as depicted in the Fig. 4
flowchart. The next iteration starts, using the updated state vector to generate new results. This process continues until pre-
defined criteria are met, in this case 30 iterations.

Figure 6 d) shows the pitch, or tangent altitude, correction factor as a function of the iterations, where it is seen to converge
after approximately 20 iterations after which further changes are minuscule. The value that the retrieval converges to is -3.269
$\pm$ 0.014 $\times 10^{-3}$ radians, which is an upward tilt of just over 0.2 degrees.

### 4.2 Trace Atmospheric Gases

Trace gas retrievals follow much the same process as the preceding pointing correction retrieval. The state vector for each
consists of the volume mixing ratio in ppmv on the defined atmospheric grid of the SASKTRAN model, which is in 1 km steps
in this case, as well as the parameters for correction of the per pixel radiometric calibration, per pixel spectral shift and the

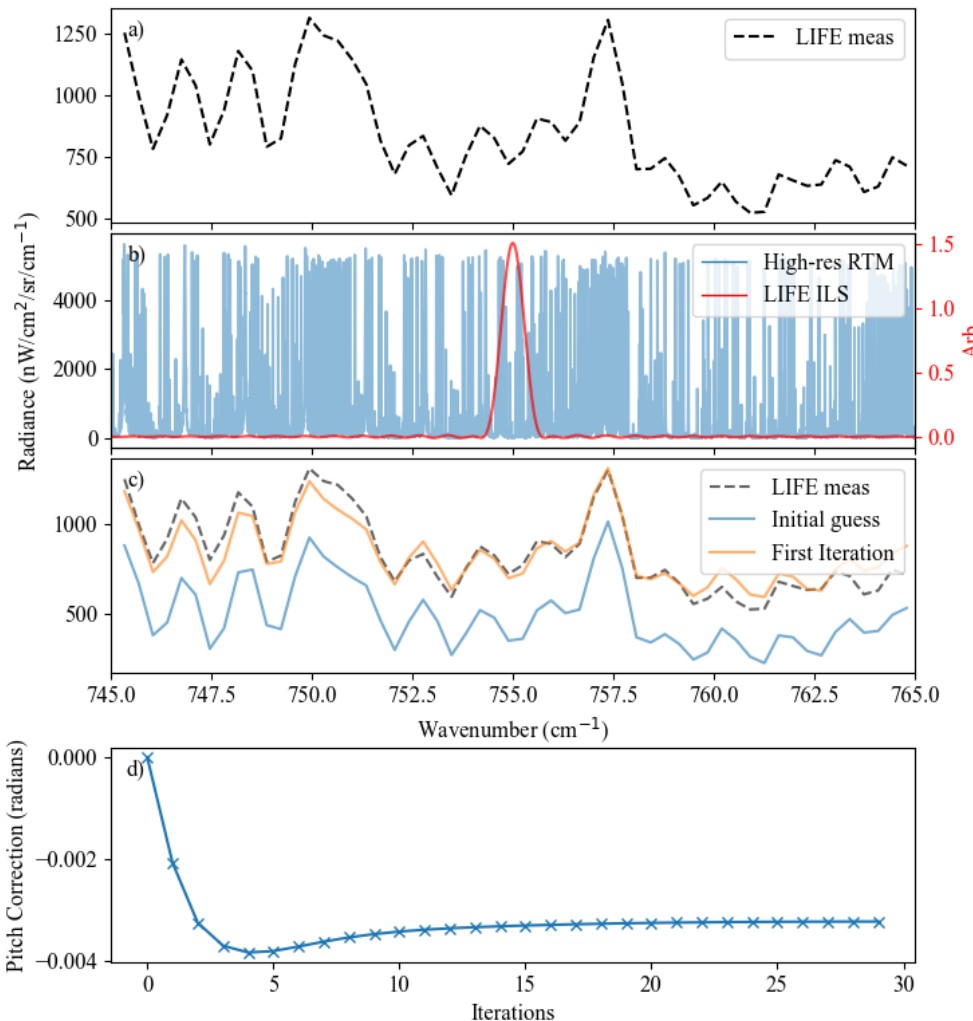

**Figure 6.** The stages of the LIFE pointing retrieval at specific points in the process. a) depicts the initial measurement for pixel 6, a pixel with LOS crossing near 27 km tangent altitude prior to correction. b) shows the high resolution result of the integration weights from the high resolution LOS grid for the pixel, overlaid with the instrument ILS. c) shows the LIFE measurements with the initial guess, before any corrections are applied, and the state of the modelled measurements after a single iteration. d) shows how the pointing correction changes as the retrieval goes through iterations. Note that a negative pointing correction corresponds to an upward tilt.

previously retrieved pointing correction. Note that the pointing correction is required in the state vector to update the states properly, but is no longer allowed to vary.

Due to the large number of unknowns in the state vector, retrieval of all parameters at once often leads to poor convergence and non-physical results. The solution is to break each species retrieval into two or three stages, an example of which is shown in Fig. 5 for the $H_2O$ retrieval specifically. The following description will use the specific example of the water retrieval to discuss the process, though each of the species has slightly different requirements.

In the first stage, the pointing previously determined is held constant while allowing the spectral shift and radiometric correction parameters to change. This can be seen as the "State Vector (a)" in Fig. 5. Taking the calibrated spectra and an unmodified atmosphere along with the state vector, the stage one retrieval seeks to find the optimal fit for broad baseline features and spectral peak locations. This result is shown as "State Vector (a) Modified" in Fig. 5. This provides a much closer starting point for the following stages, allowing the minimizer to focus on more relevant parameters. For most species, this modified state vector has the spectral shift frozen and is then used as an input for the next stage. For water however, each window needs to be treated separately for computation, leading to stage two running three times.

The second stage, for the species which require it, is an additional correction for an almost wavenumber-independent background signal in the form of a continuum. In real measurements, the baseline is increased by spectral contributions from outside the bounds of the microwindow but leave spectral peaks within the microwindow unaffected. In simulated data, the wavelengths used are only for a small spectral range outside of the microwindow and thus these effects may not be accurately represented by the model. The continuum profile aims to address this by providing the same functionality to the model with the introduction of a constant emission source over the spectral range. This continuum profile is an exponential curve as a function of altitude, multiplied by a variable scale factor that can be modified by the retrieval. In the case of the workflow of Fig. 5, continua are added as state vector parameters in each of the stage two input state vectors for all species excepting $HNO_3$, in which the correction is not necessary. Stage two allows adjustment of the radiometric correction parameters in response to changes in the applied continua, though the spectral shift is frozen to further changes, as the wavenumber scale has been accurately registered after stage one. The general process of Fig. 1 is followed in each of the stage two blocks.

The continuum profiles for each of the microwindows are added to the simulated atmosphere for stage three, excepting the $HNO_3$ case as previously mentioned. For computation speed, rather than having the continua scale factors frozen as part of the stage three state vector, the continua profiles are generated with the stage two scale factors found and then added to the modified atmosphere. Stage three follows the general process of Fig. 1 using the first stage spectral correction as a frozen parameter and the stage two radiometric corrections as the initial status of the state vector. The trace gas ppmv mixing ratio at each of the defined atmospheric grid points is finally added to the state vector as a parameter in this stage. The three stage method results in a more reasonable fit, as the amount of correction required is reduced and the adjustments to all parameters can be optimized together.

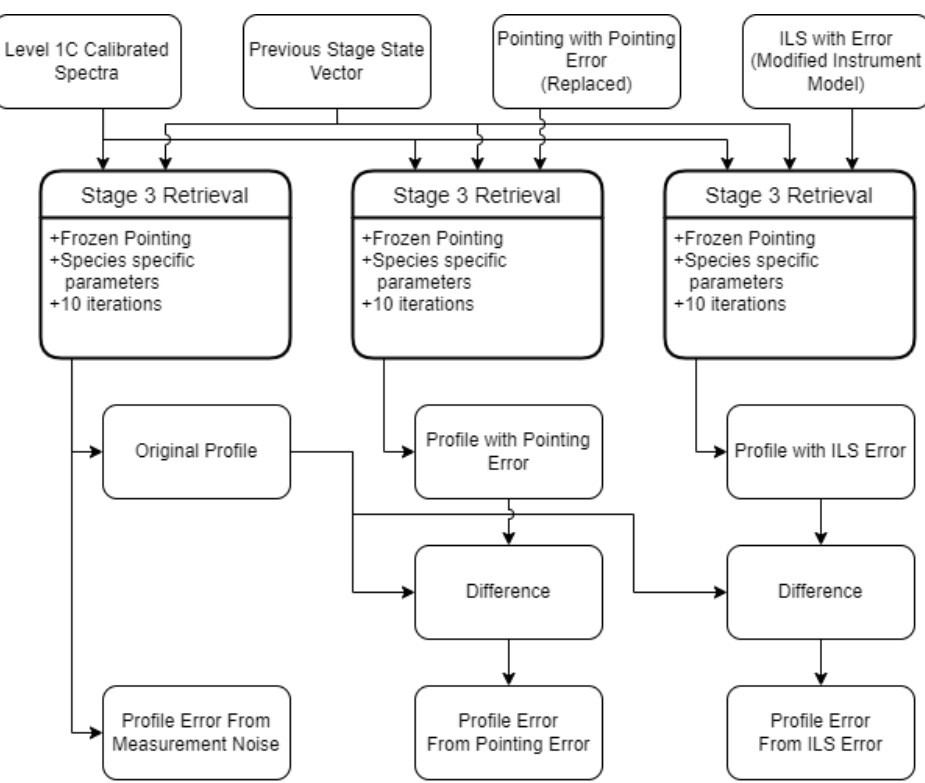

**Figure 7.** A flowchart depicting the process for determining the profile error resulting from each of the major identified error sources. This process is followed for each species. The "Stage 3" block consists of the process depicted in Fig. 1.

### 4.2.1 Error Sources

In each of the retrievals, the primary sources of error are determined. The noise error in each trace gas profile is calculated as part of the retrieval process. The other major identified sources are the pointing, uncertainty in the ILS and uncertainty in the

275 MERRA temperature profile. Figure 7 shows the process for determining the error values that are given to all species.

In the case of pointing, the retrieval outlined in Fig. 4 prescribes both a correction factor and an expected error in that value. As per Fig. 2, a small error in pointing affects all species retrievals. For each of the species, stage three is re-run with the pointing modified by the expected pointing error, which only slightly modifies the profile. The difference between this new profile and the originally retrieved profile is then considered as the error caused by the pointing correction uncertainty, which

are depicted in the central panels of Fig. 8 and 9 in yellow.

The ILS error works on the same principle as the pointing, in that a small change is made in the third stage of the retrieval and re-run to obtain a profile that includes the error. A series of measurements taken in-lab with a thermal laser indicate that the uncertainty in the FWHM of the ILS in the unapodized case is about $0.01 \text{ cm}^{-1}$. The sinc function used as the ILS in the instrument model is broadened by this amount during the new stage three profile determination. The difference between this

and the original determination is considered as the error in profile due to uncertainty in the ILS, given in green in the central column panels of Fig. 8 and 9.

The MERRA temperature profile derived from MLS measurements used in this analysis are considered to have an accuracy of 1 K or better in the altitude ranges considered by LIFE (Schwartz et al., 2008). Assuming the worst case scenario, the temperature profile is modified and all stages are re-run. The error in the profile is considered as the difference between the

290 result of the modified profile and the original. This error is given in red in the central panels of Fig. 8 and 9 and can be seen to dominate the profile error.

### 4.2.2 Single Species Retrievals Results

$O_3$ and $HNO_3$ retrievals are similar to $H_2O$ in that the spectral peaks sought within the microwindows are relatively distinct with little overlap with other species, allowing a retrieval of the species individually using the three stage methodology shown

in Fig. 5. After the pointing is optimized, water becomes the most important profile to determine, due to the broad spectral emission range. The windows used here once again are broadened versions of those used in von Clarmann et al. (2009), divided into the three major regions shown in the table in Fig. 3. The result for $H_2O$ is shown in the top two panels of Fig. 8. The black lines show the profile at the initial guess stage and after each iteration of the retrieval code. The green profile is the finalized result at the end of the tenth iteration, with a shaded green area indicating the extent of the error from all sources.

These errors are difficult to see on the scale of the profile, and are shown explicitly on a smaller scale in the center panel. The major contributor to water uncertainty is temperature uncertainty, with noise as the second largest source. The water profile converges quickly, changing minimally after the third iteration. The final water profile has a shape similar to the initially provided FASCODE profile, but shows a higher amount of oscillation. The right hand panel shows that the water retrieval over the course of time is relatively stable with no major outliers, but with the small scale structure varying from scan to scan.

Water is followed by ozone as the next major contributor in the atmosphere, shown in the second row panels of Fig. 8. Ozone makes use of the two microwindows indicated in the table in Fig. 3. As in the water case, the black lines show the result after each iteration and the green as the final result with a shaded error zone. The error source breakdown shows that the uncertainty in the vertical profile due to noise, ILS and pointing uncertainty is minimal compared to temperature uncertainty. For each of the time windows, the corresponding ozone profile replaces the standard for subsequent retrievals. The right-most panel for

ozone in Fig. 8 shows that the variation in ozone profiles, particularly at higher altitudes, seems to vary and drift, with some scans indicating a profile with less small scale structure and others having more.

The final species retrieved in the single species format is $HNO_3$. While not originally one of the science goals of the LIFE mission, the measurements show strong and distinct spectral signal in the $HNO_3$ microwindows defined in von Clarmann et al. (2009). The iterations show again that minimal changes are made beyond the third iteration, indicating a quick convergence

to the solution, and the error breakdown shows that the major uncertainty comes primarily from temperature uncertainty with measurement noise being the next major contributor. The variance seen through different retrievals from differing time periods shows that these retrievals are fairly stable over time as well.

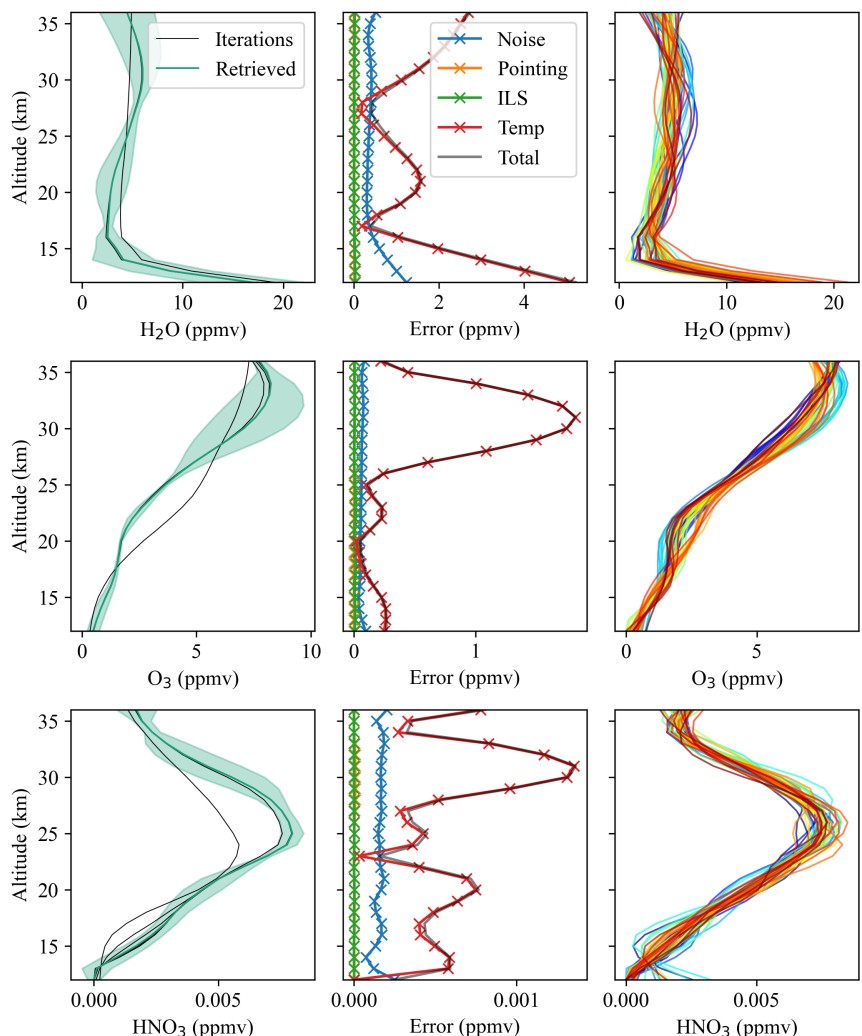

**Figure 8.** The results of the third stage for each of $H_2O$, $O_3$ and $HNO_3$ in green in the left-hand side subfigures, with each iteration shown in black and the associated errors on the center figures. The shaded regions indicate the total error from all sources around a representative retrieved profile. The right-most panel shows all retrievals over the time period 06:21:05 to 08:00:59 UTC on September 1, 2019.

### 4.2.3   Coupled Species Retrieval Results

Methane ($CH_4$) and Nitrous Oxide ($N_2O$) have significant overlap in their defined window and are best retrieved together as a joint retrieval. This methodology has been shown to work for an established source, so it is considered to be a valid approach for LIFE as well (von Clarmann et al., 2009). This joint retrieval is further complicated by a substantial systematic instrument artefact in one of the microwindows, requiring a radiometric slope correction parameter in addition to radiometric offset. Methane, in the top panel of Fig. 9 shows structure similar to ozone, where it is higher than the initial profile at high altitudes and then lower around 28 km. Nitrous Oxide shows the largest deviation from initial profile information out of any retrieved species, with the result indicating that the abundance is negligible at high altitudes before rising to a large peak at lower altitudes. These two profiles raise some interesting questions, particularly $N_2O$ with the peculiar profile shape. Considering the nature of the spectral overlap of the species with the remaining systematic error, these retrievals warrant further investigation. The plausibility of these profiles is discussed more in Section 5. The center panels of Fig. 9 show that the temperature uncertainty is once again the dominant source of error in the profiles, though the systematic error is expected to have large influence on these results with no clear method of determining the full extent. The right-hand panel shows that the stability of these results in time has a large amount of variability, especially below 25 km.

### 4.2.4   Averaging Kernels and Vertical Resolution

The atmospheric grid on which the mixing ratio for each species is determined is finer than the measurement grid, leading to a need for regularization to deal with oscillations resulting from an under-defined problem. Regularization smooths the oscillations present in such solutions at the cost of increasing the vertical resolution. Each trace gas has a different factor applied to the second-order Tihkonov regularization matrix used for regularization in a way that balances oscillation reduction with maintaining a reasonable altitude resolution. In general, the resolution after smoothing should not be much greater than the altitude spacing of the measurements, meaning that for the LIFE prototype the aim is to ensure the maximum vertical resolution remains close to 5 km when possible. The vertical resolution is determined by calculating the FWHM of the averaging kernel used in the retrieval. Fig. 10 shows the averaging kernels for each of the species in the four left-hand panels. The right-most panel of this figure shows the FWHM for each species as calculated from the averaging kernels.

The water and methane panels show much wider averaging kernels than the other species and consequently larger FWHM as a function of altitude. The reason for this is that the vertical profile retrieval of both species require more rigorous regularization. The FWHM of water peaks at just under 8 km and methane peaks at a FWHM of about 7.5 km. In each of these cases, a trade-off was made between the oscillatory nature of the retrieval and the resolution, with the current settings selected as the most appropriate compromise.

The reason for the larger regularization requirement is unclear. It may be related to the retrieval setup using all windows for all altitudes, which is not the case for the MIPAS retrievals, where certain altitude/microwindow combinations were excluded because of radiometric interference from other species (von Clarmann et al., 2009). Not selecting for specific microwindow and altitude combinations may lead to a result where the LIFE retrievals are trying to fit too many measurements and optimize

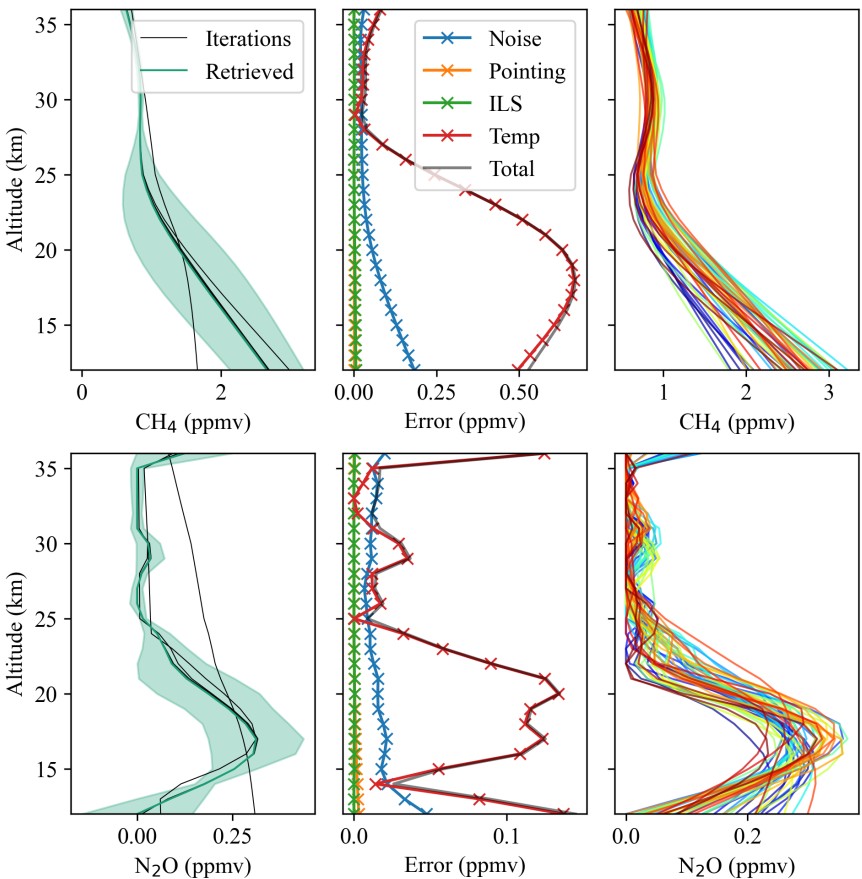

**Figure 9.** Third-stage trace gas vertical profile results for $CH_4$ and $N_2O$, which were co-retrieved, along with their major error contributors. The shaded regions indicate the total error from all sources around a representative retrieved profile. The right-most panel shows all retrievals over the time period 06:21:05 to 08:00:59 UTC on September 1, 2019.

inefficiently. The use of particular microwindows for different altitudes is identified as an area of further exploration in the LIFE instrument and its descendants.

The remaining species, $O_3$, $HNO_3$ and $N_2O$, are shown to each have much smaller FWHM values and better vertical resolution with a value at or below 5 km for the entire range with the exception of ozone which peaks at around 6 km at upper altitudes.

## 5 Comparisons to ACE and MLS

Observations obtained with the ACE-FTS instrument several days after the LIFE demonstration flight and MLS observations on the flight day are plotted with the corresponding LIFE observations in Fig. 11. Nine ACE-FTS profiles that cover a time

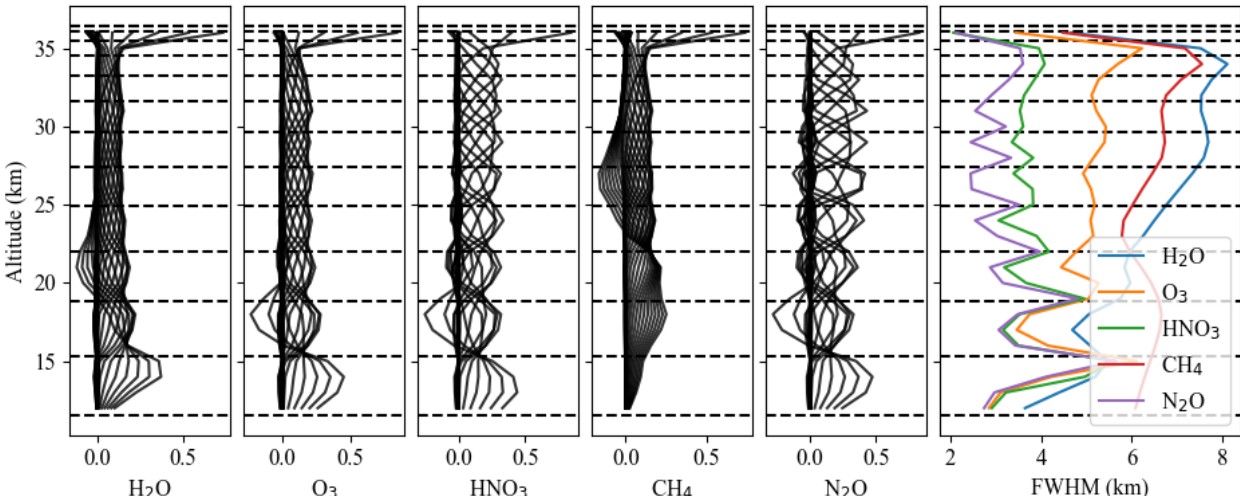

**Figure 10.** The averaging kernels for each of the LIFE retrieval species and the overplot of the vertical resolution for each, determined by the FWHM of the averaging kernels for each altitude grid-point.

range from September 6 to 9, 2019, with a longitude range from -56 to -105 degrees and a latitude range from 43 to 52 degrees are used in this comparison. Three MLS profiles from September 1, 2019, with a longitude range of -75 to -77 degrees and a latitude range of 47 to 51 degrees are also used in the comparison. The Timmins flight took place from August 31, 2:00 UTC until September 1, 16:45 UTC in 2019, with the specific retrievals used in this paper being derived from measurements taking place between 06:21:05 and 08:00:59 UTC. The general location during flight was about 48 degrees latitude and -81 degrees longitude.

For the sake of comparison, the mean and standard deviation of the LIFE retrievals have been found and plotted alongside the ACE and MLS profiles. Figure 11 shows the side-by-side comparison of the mean profile for a given species (dark green line) with an error (shaded green area) defined by the standard deviation, the ACE counterparts given in dashed blue and MLS measurements in red. From left to right, the species are $H_2O$, $O_3$, $HNO_3$, $CH_4$ and $N_2O$.

In the case of $H_2O$ and $O_3$, the mean retrieval appears to have a low bias, especially at the higher altitudes, with the standard deviation not quite overlapping with the comparison profiles, seeming to indicate a low bias in these atmospheric ranges. Ozone transitions to a high bias below 20 km altitude as well, which appears unique amongst the retrievals. The $HNO_3$ retrieval also appears to underestimate the peak at 25 km. The $CH_4$ and $N_2O$ retrievals show the most deviation from the ACE measurements. For $CH_4$, there are some altitude regions for which the LIFE retrievals and ACE do not overlap, in the region where LIFE appears to have a low bias in the other species and in the region below the tropopause, which can be somewhat expected from the atmospheric variability in that region. $N_2O$ shows less agreement, with a large bias towards lower values in the 20 to 30 km range before peaking at lower altitudes. It is likely that remaining systematic biases in the spectral region and the need for joint retrievals are exacerbating factors. The current version of the instrument clearly struggles with $N_2O$

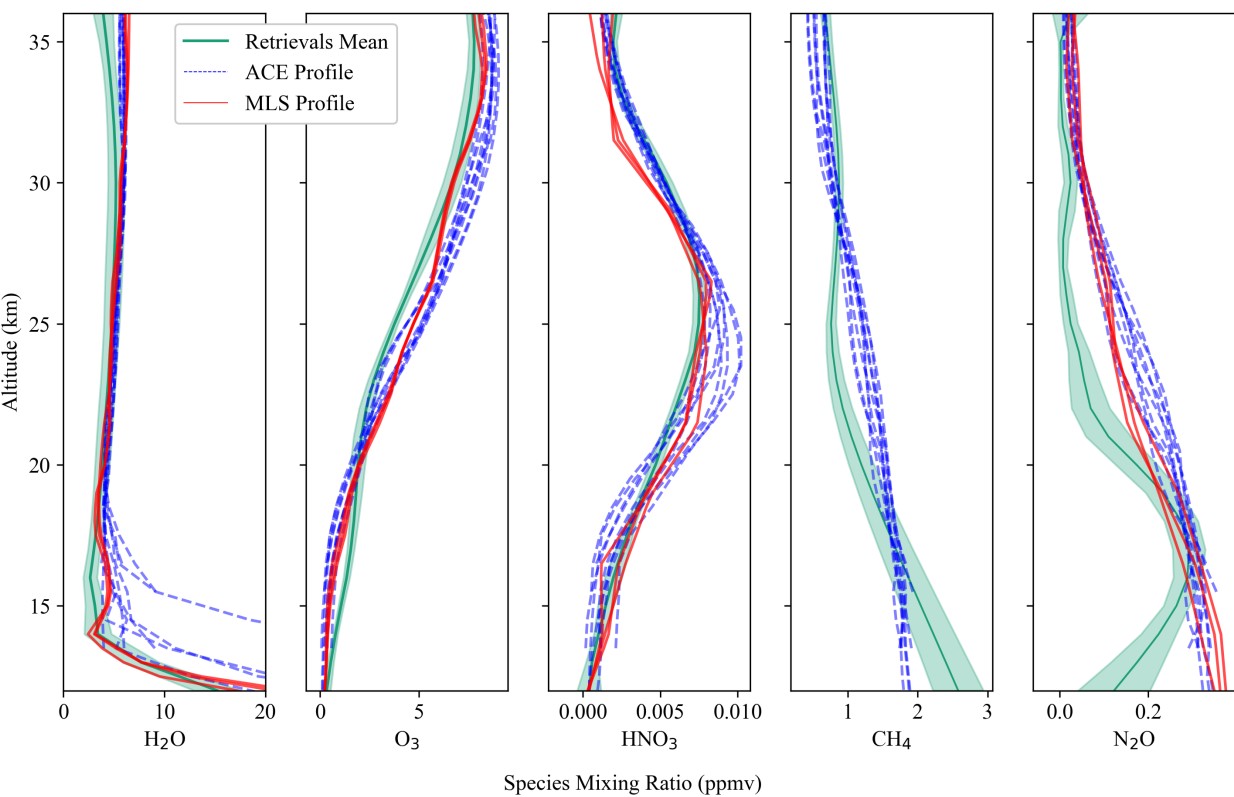

**Figure 11.** Comparison of each of the LIFE target trace gas average vertical profiles retrieved (green) on September 1, 2019 to the same species profiles as determined by the ACE instrument measurements (blue) in the time span of September 6 to 9, 2019, longitude range -56 to -105 degrees and latitude range 43 to 52 degrees and as determined by MLS observations (red) from September 1, 2019, longitude range -75 to -77 degrees and latitude range 47 to 51 (excluding $CH_4$).

retrievals and with CH$_4$ to a lesser degree, but these results indicate that accurate profiles are a possibility if systematic biases can be removed. It should be noted that the closest ACE profile available was September 6, five days after the Timmins flight,

which could also account for some of the differences seen in all of the species. In general, the LIFE measurements show closer agreement with the MLS observations, though the same biases exist. A large part of this could be attributed to the temperature error in the profiles, as all species have high temperature dependence.

While there is not perfect agreement, the LIFE profiles for H$_2$O, O$_3$ and HNO$_3$ exhibit agreement in both shape and magnitude with the other instruments. The cause of the larger observed biases between the CH$_4$ and N$_2$O profiles remains unknown;

residual errors in the radiometric correction in the microwindows is a possibility. The standard deviation exhibited across all retrievals is indicative of both instrument parameters and atmospheric variability over the course of the retrievals that are not fully understood and accounted for.

## 6 Conclusions

Instruments that make use of IFTS technology have been identified as a promising candidate for furthering the understanding

of the UTLS region of the atmosphere. The LIFE prototype instrument successfully demonstrates how the methodology can be implemented with commercially available off-the-shelf components with reduced complexity while still obtaining reliable trace gas retrievals. The successful flight in Timmins in August 2019 demonstrates that such an instrument is capable of being flown in the near-space environment. The atmospheric trace gas profiles obtained and their similarity to ACE and MLS measurements taken at a similar time and place indicate that the data taken from such a platform also provides reliable scientific data. For these

reasons, the pursuit of further instrumentation aboard a satellite platform and the creation of a LIFE successor is justifiable.

The major weakness identified with the trace gas retrievals is the need for further radiometric correction. Originally identified in the companion publication detailing the instrument characterization, artefacts that remain after the initial radiometric correction required additional steps to correct and are a likely source of error in the trace gas profiles obtained that are not encapsulated by the expected instrument error. The blackbodies used in the calibration during flight are identified as the cause

of radiometric calibration error, as they were not originally purpose-built for the LIFE instrument. Another consideration is that a deep space measurement is helpful in identifying and eliminating non-linear effects on the radiometric calibration, which the prototype was unable to take. Future iterations taking these factors into account with custom modifications will avoid the issues encountered with the LIFE prototype.

The scientific goal of the prototype LIFE instrument was to demonstrate that commercially available IFTS technology

with reduced thermo-optic complexity produces valid atmospheric measurements that can be used for atmospheric trace gas profile retrievals. The goal for validation identified H$_2$O, O$_3$, CH$_4$ and N$_2$O as the greenhouse gases most important to the understanding of the UTLS region while also being relatively straightforward for the instrument to measure. Based on the demonstration flight, HNO$_3$ was added due to its strong signal, and though not included in this publication, there is a potential for further retrievals of other atmospheric constituents. The presented results indicate that LIFE exceeded the initial goals,

demonstrating the feasibility of a satellite-borne version of an instrument using the same methodology.

*Author contributions.* ER was responsible for Formal analysis, Investigation, Software and Writing the original draft. JL was responsible for Investigation. DZ was responsible for Methodology. AB was responsible for Methodology and Conceptualization. DD was responsible for Conceptualization. All listed authors were responsible for the review and editing process.

*Competing interests.* The authors declare that they have no conflict of interest.

*Acknowledgements.* This study was partly funded through the FAST initiative of the CSA. The study was also made possible by collaboration with ABB Canada through the provision of components.

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
