# Peer review of "Stratospheric trace gas profile retrievals from balloon-borne limb imaging of mid-infrared emission spectra"

_Atmospheric Measurement Techniques, 2023_

## Author Comment (AC2)

**Runge et al. AMT-2023-9: Author Response**

Thank you to the reviewers for their positive comments and constructive feedback. Our response to the reviewer comments are provided in red below.

**Reviewer #1**

The manuscript has already been in a good shape when I provided my access review. The authors have carefully considered my few comments made then, thus I do not have much to add and suggest publication of the paper as is. With respect to the high bias of methane, the authors may speculate that inconsistence in the spectroscopic data in the spectral region used by LIFE and in the spectral region used by ACE-FTS might be part of the explanation.

A discussion of the reliability of the LIFE retrievals in light of MLS measurements and a discussion of the systematic in the CH4/N2O window is added for clarification.

**Reviewer #2**

**General comments:**

The authors present results of a demonstration flight of the imaging Fourier transform spectrometer LIFE on a stratospheric balloon. After a comprehensive description of the data processing, the authors present vertical profiles of the trace gases $H_2O$, $O_3$, $HNO_3$, $CH_4$, and $N_2O$. These profiles are compared to observations of the ACE-FTS satellite instrument. The manuscript is clearly written and well-structured and will be of interest to scientists involved in projects with imaging spectrometers. I therefore recommend publishing this manuscript in AMT after addressing the comments below.

**Major comment:**

My main concern affects the comparison of the trace gas profiles measured with LIFE and obtained with ACE (since the authors consider this comparison as a kind of validation of the LIFE measurements). The differences in measured vertical trace gas profiles (mainly $N_2O$ and $CH_4$, but also $H_2O$ and $O_3$) between both instruments are quite large in some altitude regions, exceeding the combined errors of both instruments. Some differences may be explained because the temperature was not fitted prior to the LIFE trace gas retrievals. Another reason may be the poor coincidence between LIFE and ACE observations since the ACE measurements were recorded several days later than the LIFE measurements. I therefore strongly recommend that the authors also compare Microwave Limb Sounder (MLS) profiles that should be available for the gases $H_2O$, $O_3$, $HNO_3$, and $N_2O$ close to the time and the region of the LIFE measurements and plot them as an additional mean profile (including standard deviation) in Figure 11.

Both temperature error estimation and MLS profiles have been added to the analysis of the LIFE data as per the reviewer's feedback

**Specific comments:**

Line 105 and Figure 2: You should add one or two sentences how you treated the temperature during the retrieval process. Obviously, the temperature profile was not retrieved and you used a meteorological temperature analysis instead. Since the assumed temperature has some influence on the trace gas retrieval results, you should explain this matter clearly.

A statement has been added addressing that a MERRA2 temperature profile is used rather than one determined by the LIFE instrument. A statement indicating that the uncertainty resulting from this decision is discussed in Section 4.2.1 is added as well.

Page 9, Figure 5, Box "Stage 2": What means "Low only" here? Does this mean that the continuum is only taken into account at low altitudes? Please clarify.

This was an error left from an earlier version that had multiple windows. "Low only" was referring to the use of the spectral window with lowest spectral number. The figure has been updated to omit this statement as it is not relevant.

Line 213 and Figure 6b: It is not clear to me what is meant with "high resolution" here. It is also not clear to me what exactly is plotted as high-res RTM in Figure 6b. Is it important to show this parameter? If yes, please explain this a bit more in detail. If no, please omit this parameter in the plot.

Clarification that the blue plot in Figure 6 b) is an example output from the radiative transfer model along the LIFE line of sight prior to convolution with the instrument line shape has been added

Lines 260-262 and Figures 8 and 9: You mention some "primary sources" of retrieval errors. However, from my experience inaccuracies in the temperature profile used for the trace gas retrievals is also a major error source, especially if the temperature profile is not retrieved prior to the trace gas retrievals. You should include at least one or two sentences here why you obviously neglected the temperature error. The better way would be to estimate the temperature error by test retrievals and to include this error in Figures 8 and 9.

An estimation of the temperature error has been added to the discussion. You are correct in your assessment that this is the most significant source of error in the retrievals and the paper is updated to reflect this.

Line 309: "While the shape of these profiles raises interesting questions for further investigations, the magnitudes recovered are plausible ...". That sounds very vague. The $N_2O$ magnitudes are only plausible between about 15 and 22 km. Above this altitude region they would only be plausible within the late winter polar vortex but not at mid-latitudes in summer. Below about 15 km, the $N_2O$ profile is completely unreasonable. Interesting questions are, for instance: Is an inaccurate temperature profile at least partly responsible for the extremely low $N_2O$ values? Are the selected microwindows strongly influenced by overlapping features of different gases due to the coarse spectral resolution (at least with regard to $N_2O$ because the retrieval seems to work better for $CH_4$)? Please be more specific and write a few sentences about possible reasons for this strange $N_2O$ profile.

A brief discussion around the N2O profile being implausible is added, along with a more direct and immediate reference to the systematic error in the spectral range used in the N2O retrieval.

Lines 349 and 350: For ozone, the mentioned low bias is only visible above 20 km. Below this altitude, the bias is positive.

The text has been updated to clarify this.

Line 352: "...there are only a few regions for which the retrievals and ACE...". It should read like "...there are some altitude regions for which the LIFE retrievals and ACE...".

The wording has been updated to reflect this correction.

Lines 353 and 354: "...from the turbulent nature of the atmosphere...",. You probably mean "...from the atmospheric variability...".

This wording is more appropriate, the paper has been updated accordingly.

Lines 355 to 360: There is a lot of speculation here about the differences between LIFE and ACE. As mentioned at the beginning, for reasons of a better coincidence between LIFE and a comparison instrument, MLS trace gas profiles should also be shown in Figure 11, as these are available for all gases displayed (except $CH_4$) for the day of the flight. The authors could chose mean MLS profiles around Timmins (e.g. +/-5 deg. latitude +/-20 deg. longitude) including standard deviation. The present text should then be adapted accordingly (here and in the conclusions around line 366).

Three MLS profiles from the flight day, September 1, have also been added to the figure for comparison. This allows for some slightly better agreement between LIFE and an established instrument, but the biases discussed in regard to the ACE

measurements are still seen to a certain degree. MLS also indicates the N2O retrieval is poor. The manuscript is updated to discuss these points with the addition of MLS profiles.

Lines 364, 367, and 378: The word "valid" sounds too strong; this reminds me of a valid law, which is not the case here. You may write instead "reliable" or "confidable".

The language has been changed to "reliable" rather than "valid" in these cases.

Line 382: "...other atmospheric constituents appear promising as well". This is a very vague statement. Which species are meant here, for example? The expected accuracy is likely to be less than that of the gases previously discussed. Please be a little more specific in your statements here.

The statement has been changed to indicate that there is a potential to recover more species than the five from this paper. A proper study of which species could be recovered and how reliably is beyond the scope of this publication, and the new statement is more in keeping with this idea.

Line 383: "...validity of a ...". This statement is again too strong. Please change to "...feasibility of a ...".

This change is accepted.

**Technical corrections:**

Line 13: Please define acronym ABB.

ABB Canada is the name of the company, not an abbreviation of the company name, so this is left as is.

Line 20: Please define acronym CNES.

Accepted, completed.

Line 26: "... designed for an aircraft ...". Please rewrite "... designed for aircraft and balloon platforms ...".

Accepted, completed.

Line 29: Please include the paper by Johansson et al. (Atmos. Meas. Tech., 11, 4737–4756, doi:10.5194/amt-11-4737-2018, 2018) into this list since this paper contains the validation aspects mentioned in line 28.

This is a good reference for this work, thank you for the suggestion.

Line 51: Please write "follows" (instead of "follow").

Accepted, completed.

Page 4, Fig. caption 1, line 4: Please change …" is output as the result" into "… output is the result".

Accepted, completed.

Line 123: Please define acronym FASCODE.

Accepted, completed.

Line 124: Please define acronym MERRA.

Accepted, completed.

Line 209: Please define acronym NOAA.

Accepted, completed.

Line 211: "…is applied is shown…". Please rewrite "…is applied and is shown…".

Accepted, completed.

Page 15, Fig. caption 8 and Page 16, Fig. caption 9: Instead of writing "…on the flight date" please give the exact date of the flight here. Please indicate the meaning of the shaded region in the left hand panel.

The date of the flight has been added, as well as an indication of the meaning of the shaded regions.

Page 18, Fig. caption 11: "…from around the same time and location". Please be more specific and give the time difference in days and the maximum spatial distance.

The figure caption has been updated to be more specific with the time, latitude and longitude range for both the ACE and MLS profiles used for comparison.